# Lactoferrin, Quercetin, and Hydroxyapatite Act Synergistically against *Pseudomonas fluorescens*

**DOI:** 10.3390/ijms22179247

**Published:** 2021-08-26

**Authors:** Angela Michela Immacolata Montone, Marina Papaianni, Francesca Malvano, Federico Capuano, Rosanna Capparelli, Donatella Albanese

**Affiliations:** 1Department of Industrial Engineering, University of Salerno, 84084 Salerno, Italy; angela.montone@izsmportici.it (A.M.I.M.); fmalvano@unisa.it (F.M.); dalbanese@unisa.it (D.A.); 2Department of Food Inspection, Istituto Zooprofilattico Sperimentale del Mezzogiorno, 80055 Naples, Italy; federico.capuano@cert.izsmportici.it; 3Department of Agriculture, University of Naples “Federico II”, 80055 Naples, Italy; marina.papaianni@unina.it

**Keywords:** *Pseudomonas fluorescens*, HA–Lacto–Que complex, antimicrobial additives

## Abstract

*Pseudomonas fluorescens* is an opportunistic, psychotropic pathogen that can live in different environments, such as plant, soil, or water surfaces, and it is associated with food spoilage. Bioactive compounds can be used as antimicrobials and can be added into packaging systems. Quercetin and lactoferrin are the best candidates for the development of a complex of the two molecules absorbed on bio combability structure as hydroxyapatite. The minimum inhibiting concentration (MIC) of single components and of the complex dropped down the single MIC value against *Pseudomonas fluorescens.* Characterization analysis of the complex was performed by means SEM and zeta-potential analysis. Then, the synergistic activity (*C_syn_*) of single components and the complex was calculated. Finally, the synergistic activity was confirmed, testing in vitro its anti-inflammatory activity on U937 macrophage-like human cell line. In conclusion, the peculiarity of our study consists of optimizing the specific propriety of each component: the affinity of lactoferrin for LPS; that of quercetin for the bacterial membrane. These proprieties make the complex a good candidate in food industry as antimicrobial compounds, and as functional food.

## 1. Introduction

*Pseudomonas fluorescens* is a Gram-negative, aerophile, and psychotropic bacterium. The psychotropic property of this bacterium favors its growth—even in cold rooms—and reduces the shelf life of food products. Thus, *Pseudomonas fluorescens* contributes particularly to the deterioration of fish, milk, and meat stored at low temperatures.

*Pseudomonas* spp.—including *Pseudomonas fluorescens*—account for up to 97% of bacteria isolated in cattle meat stored in cold rooms [1,2,3,4]. These species secrete several enzymes (protease, lipase, and thermostable lecithinase) that cause deterioration of food products [5,6], a source of heavy economic losses to the food industry [7]. So far, numerous antimicrobial additives have been exploited in order to control microbial contamination during the food supply chain [8,9,10]. Generally, consumers consider food additives as very unhealthy. Since consumers see chemical additives as dangerous, the food industry has decided to use natural molecules to control bacterial contamination [11,12,13]. Natural molecules often are active only at high concentrations and against a limited number of bacterial species. This limit can partially be reduced using lower concentration of several components. The food industry generally prefers adding additives to the packaging, rather than directly to the product [14]. However, this solution needs using a carrier to control the additive release and a scaffold to preserve the activity of the additives throughout the shelf life of the product.

Among the potential components to use against *Pseudomonas fluorescens*, we exploited quercetin, lactoferrin, and hydroxyapatite. Quercetin (que) is a polyphenolic molecule common in plants and known as an efficient antioxidant. In plants, quercetin is present in the forms of quercetin-3-O-rutinoside (rutin), quercetin-3-O-glycoside, and quercetin-3-O-(6″-O-acetil)-glycoside. Quercetin exerts its antioxidant activity principally by eliminating free radicals [15], chelating metallic ions [16], and inhibiting lipid peroxidation [17]; in addition, it is active against bacteria and yeast (*Pseudomonas aeruginosa*, *Salmonella enteritidis*, *Staphylococcus aureus*, *Escherichia coli*, *Proteus* spp., and *Aspergillus flavus*) [18]. 

Quercetin exerts its antibacterial activity by degrading the bacterial cell wall, altering cell permeability, protein synthesis, enzymatic activity, and inhibition of nucleic acid synthesis [18,19,20,21,22]. 

Lactoferrin (lacto) is a glycoprotein with a molecular weight of 78 Kd. Its peculiar property is of binding iron—abundant in animal and human milk—and in exocrine secretions. Originally, the antimicrobial activity of lactoferrin was attributed to iron chelation; at present, we know that it directly damages the external membrane of Gram-negative bacteria binding to the lipid A of LPS (Lipopolysaccharide) and promotes its removal from the bacterial surface [23]. Further, lactoferrin and quercetin are both anti-inflammatory molecules [24]. In particular, quercetin controls the activation of *NF-kB* (nuclear factor kappa-light-chain-enhancer of activated B cells) [25] and the release of pro-inflammatory cytokine (*TNF-α*, *IL-1β*, *IL-6*, *IL-8*) [26]. Lactoferrin modulates the expression of pro-inflammatory cytokine [27,28], chemiotactic factor production, and the expression of adhesion molecules [29]. This property as suggested their use against COVID-19 [30]. 

Hydroxyapatite (HA) [Ca_5_(PO_4_)_3_(OH)]—an inorganic mineral and basic component of the bones—has also been shown efficient as a carrier-molecule for its property of interacting with antimicrobial peptides, and organisms such as bacteriophages [31,32]. Moreover, it has already been successfully tested as a stabilizer of quercetin [33] and lactoferrin [34] against *Pseudomonas* spp. and *Salmonella* spp., respectively.

The purpose of the present study was to test the potential utility of a complex that included quercetin and lactoferrin as antimicrobials, and hydroxyapatite as a carrier for the food industry.

First, we tested in vitro the antimicrobial activity of quercetin and lactoferrin against *Pseudomonas fluorescens*, individually and in combination as a complex with hydroxyapatite. We established the minimum inhibiting concentration (MIC) of single components and of the complex. Then, we calculated the synergistic activity (*C_syn_*) of single components and of the complex. Finally, we confirmed the synergistic activity of the complex, testing in vitro its anti-inflammatory activity on U937 macrophage-like human cell line.

## 2. Results

### 2.1. Antimicrobial Activity of Que and Lacto Alone and Complexed with HA

First, we tested separately Que and Lacto against *Pseudomonas fluorescens*. Que displayed a slightly higher antimicrobial activity (Figure 1a). Next, we tested Que and lacto, each in combination with HA. Que and Lacto displayed the same MIC values (500 ppm) (Figure 1b), which is much higher, compared to the concentration of 250 ppm reported as the maximum concentration approved [29,35], suggesting that HA does not show a significant improvement of Que and Lacto antimicrobial activity.

In order to determine their optimal concentrations, Que and Lacto were tested together against *Pseudomonas fluorescens*. The MIC of the two molecules together was 200 ppm (*w*/*v*) (Figure 2). This MIC value (200 ppm) was combined with HA and tested at different concentrations and different combinations. Specifically, interaction of Que and Lacto with HA displayed different antibacterial activity, depending upon which molecule was added first to HA. The highest inhibition was obtained when the HA was first incubated with Lacto and then with Que (smaller compared to lacto) at a concentration of 100 ppm (*w*/*v*) (Figure 3b). Instead, when the order was inverted (HA–Que–Lacto) at the same concentration (100 ppm), the inhibition was only 40% (Figure 3a).

### 2.2. Evaluation of HA Absorption Capacity

Results of the absorption showed that Que was fully absorbed on HA structure (1:100 *w*/*v*) for all quantities of Que tested (50–200 ppm) (Figure 4a). Additionally, for Lacto the absorption on HA is total, even between 50 and 100 ppm the absorbance results in negative values (Figure 4b). The orange points in both graphs correspond to supernatant’s absorption after reaction with HA.

### 2.3. Synergistic Activity of the Complex

The complex HA–Lacto–Que, at the concentration of 100 ppm (*w*/*v*)—alone—can inhibit Pseudomonas fluorescens (Figure 3b). Next, we looked for a synergistic activity of the complex. The results shown in Table 1 clearly demonstrate that the highest activity occurs when the complex HA–Lacto is used with Que at 100 ppm (*C_syn_*= 6.75 ± 2.67). However, the synergistic effect is also observed when the Que was used at 200 or 50 ppm (suboptimal concentrations). The fractional inhibitory concentration index (FIC index) was calculated to confirm the synergistic effect as reported by Bidaud et al. [36]. The complex HA–Lacto–Que, at the concentration of 100 ppm (*w*/*v*) has a synergistic effect with a value of 0.4 according to the interpretation that a FIC index of ≤0.5 suggests the synergistic interaction.

### 2.4. Characterization of the HA–Lacto Complex in Presence of Que 

To examine the interaction of HA with Lacto and Que at molecular level, we used the scanning electron microscope (SEM). HA show porous spherical aggregates of elongated crystallites. The average dimension of the particles was above the expected size of nanometer. This finding may result from the self-aggregation of the HA from a few hundred nanometers to a few microns (Figure 5a) this hypothesis is in line with the absence of net charge of zeta potential [37]. The functionalization of HA with Lacto alone (Figure 5b) and with Lacto–Que (Figure 5c) reduced the porosity of HA as effect of adsorption of the two compounds inside the crystalline structure.

This result is confirmed by independent z-potential data. In Figure 6 the HA–Lacto, with a z potential of −30 mV compared to HA alone, with values of −15 mV, show a strong aggregation of HA nanocrystals. The HA–Lacto–Que complex shows a positive z potential (11 mV), which demonstrates that the positive electrostatic surface potential of Que produces a strong surface interaction with the HA nanocrystals and stabilizes the previous HA–Lacto bond, which suggests less repulsion between the components of the complex [14].

### 2.5. Anti-Inflammatory Activity of the Complex

The human macrophage cell line U93 upon incubation after 6 h with *Pseudomonas fluorescens* displayed activation of the pro-inflammatory cytokines *TNF-α*, *IL6*, and *IL8* (Figure 7). Instead, when the same cell line was incubated with the complexes or the single components, we observed a significant reduction of cytokine production. The experiment confirms, once more, the synergistic activity of the complex (HA–Lacto–Que). *IL8* is the only cytokine that shows a marked difference in the presence of Lacto (4.08 Fc) or Que (11.74 Fc). In other words, *IL8* is down regulated when Que is more exposed on a complex surface.

## 3. Discussion

*Pseudomonas fluorescens* is an opportunistic, psychotropic pathogen that can live in different environments, such as plant, soil, or water surfaces. *Pseudomonas fluorescens* has enzymes acting on proteins, lecithin, and lipids conferring to the aliment an undesirable taste [38]. Further, *Pseudomonas fluorescens* damages several plants causing chlorotic and necrotic lesions on leaves and fruits, with heavy economic losses to the agriculture and the food industry. The chemical additives, at present used to preserve the food against the pathogens, are perceived unhealthy by the consumer. 

This study provides clear evidence that the bovine lactoferrin and the quercetin glycoside act against *Pseudomonas fluorescens* efficiently. Provided that data reported in literature are generally difficult to compare, our results (antimicrobial activities of lactoferrin and quercetin equal or up to 300 ppm) (Figure 1a) agree with the literature [39,40]. However, when the two molecules were combined, the MIC dropped to 200 ppm (Figure 2).

In the next step, the two molecules (Lacto and que) were combined with HA. The MIC dropped further down to 100 ppm (Figure 3a). The best results were obtained when Lacto was added first to HA (Figure 3b). One more positive feature emerging from our data is that Que and Lacto are completely adsorbed to HA (Figure 4a,b) and the complex is stable, with a z-potential value of 11 mV (Figure 6). The z-potential measures the repulsive forces between particles: the higher is the repulsive force, the lower is the probability of aggregate formation. Moreover, to assess if the effect induced by the simultaneous treatment with the HA–Lacto–Que and HA–Que–Lacto were additive or synergistic were analyzed two parameters: normalization as a function of control absorbance (ρ); and synergistic coefficient (*C_syn_*). Since high *C_syn_* values indicate a clear synergistic effect, the best result was obtained when the antimicrobial treatment was performed using HA–Lacto–Que complex (Table 1).

*Pseudomonas fluorescens* is generally considered as non-pathogenic for humans. However, this pathogen has been detected in human clinical samples and shown to be highly hemolytic and able to induce cytotoxic and pro-inflammatory response [41]. An antigen from the same pathogen has been isolated from the serum of patients with Crohn’s disease [42].

Back to our study, in order to evaluate in vitro the anti-inflammatory activity of the complex, we infect the human macrophage cell line U937 with *Pseudomonas fluorescens*. We noticed a statistically significant reduction of several pro-inflammatory cytokines (Figure 7). The high level of *IL8* may be assigned to the presence of intracellular bacteria not killed by the Lacto (known to have a moderate antimicrobial activity) [43]. In the presence of que, *IL8* is down regulated. This result is in line with the known activation of IL8 through the TLR4-independent pathway [44]. From this result, we can also infer that Que is not induced by LPS, a TLR4-dependent pathway. Consequently, the down regulation of *IL8* reported above can appropriately be attributed to the presence of Que exposed on the complex (HA–Lacto–Que).

In conclusion, the peculiarity of our study consists of optimizing the specific propriety of each component: the affinity of Lacto for LPS; that of Que for the bacterial membrane, a property that we amplified by adding Que to the complex last and, therefore, more exposed (Figure 8). These properties make the complex as good candidate for antimicrobial use in the food industry, and as functional food.

## 4. Materials and Methods

### 4.1. Bacterial Strains and Culture Conditions

The bacterial strains of *Pseudomonas fluorescens* strain ATCC 13525 were provided from the “Istituto Zooprofilattico Sperimentale del Mezzogiorno” in Portici (Naples, Italy). *Pseudomonas fluorescens* was grown overnight at 37 °C in the liquid culture medium (Buffered peptone water, BPW). To identify bacterial growth phase, turbidity of medium was measured by optical density measurement at 600 nm on a UV/Vis spectrophotometer.

### 4.2. Antimicrobial Compounds

Quercetin glycoside compound (98.6% food grade) was purchased from Oxford^®^ Vitality Company (Bicester, UK) and lactoferrin (95% food grade) from Fagron the UK. The stock solution of each compound was dissolved in water, to obtained a final concentration of 1 mg/mL. 

### 4.3. Antimicrobial Activity

The minimal inhibitory concentration (MIC) of each compound was determined by a colorimetric method, using 3-4,5-dimethylthiazol 2,5-diphenyltetrazolium bromide solution (MTT), using the standard broth microdilution method according to [33]. All antimicrobial tests performed in this work were carried out using the same method.

### 4.4. Biomimetic HA Nanocrystal Synthesis and Characterization

Biomimetic hydroxyapatite nanocrystals were produced as described by [45]. HA was precipitated from a solution of (CH_3_COO)_2_Ca (75 mM) by the slow addition of an aqueous solution of H_3_PO_4_ (50 mM) and maintaining pH at 10 (addition of NH_4_OH). The synthesis was carried out at room temperature (RT). Finally, the suspension of HA was washed with distilled water to remove ammonium ions and favor the interaction between nanocrystals.

### 4.5. Evaluation of HA Absorption Capacity and Complex Preparation

HA–Que and HA–Lacto complexes were prepared adding HA solution diluted (1:100) (4% *w*/*v*) with known concentrations (500, 400, 300, and 200 ppm) of Que or Lacto, respectively. The complexes were gently mixed at room temperature for 24 h. After the incubation, the solutions were centrifuge and the supernatants and the effective amount of Que and Lacto entrapped into the structure were measured. Specifically, the amount of Que was evaluated using UV-vis spectrophotometer (Perkin Elmer Lambda 25) at a wavelength of 369 nm; the standard curve consisted of 10–200 ppm Quercetin-3 glucoside. The amount of Lacto was evaluated using spectrophotometer Nanodrop ONE_C_ at a wavelength of 280 nm and the standard curve consisted of 10–500 ppm BSA (Bovine Serum Albumin). Finally, the complexes HA–Que and HA–Lacto were vacuum evaporated for 2 h, then the second molecule—Que or Lacto (2000–100–50 ppm)—was added; finally, the complexes were mixed for 24 h and their antimicrobial activity was evaluated.

### 4.6. Statistical Analysis for the Complex Synergy

The additive or synergistic interaction of each complex was calculated using the parameters ρ (normalization as a function of control absorbance) and *Csyn* (synergistic coefficient) statistical approach:

Once defined, the ρ parameter was:ρa = 1 − (abs_ctrl_ − abs_a_)/abs_ctrl_(1)

It was possible to estimate the synergistic coefficient *Csyn* as:*Csyn* = ρ_a_ ∗ ρ_b_/ρ_ab_(2)

Error analysis was performed according to [46].

Moreover, the FIC index can be used to confirm the effect of a tested combination. The FIC is designed by division of the MIC of the complex and the MIC of the molecules alone according to the following formula:
FIC index = FIC A + FIC B = (MIC complex1/MIC 1 alone) + (MIC complex2/MIC 2 alone).(3)

### 4.7. Zeta Potential

The dimensions of HA, HA–Lacto, and HA–Lacto–Que were measured with the zeta potential using a Zetasizer Nano ZS (Malvern Instruments, DTS1070, Malvern, UK). Each sample was tested in triplicate using 1 mL of sample at 25 °C.

### 4.8. SEM Image

Water suspensions of HA, HA–Que, HA–Lacto, and HA–Lacto–Que samples were centrifuged at 13,000 rpm for 15 min and then deposited on 5 × 5 mm silicon chips; the solvent was evaporated under vacuum at 30 °C; the silicon supports were mounted as described in [46]. SEM microscopy was recorded with a NovaNanoSem 450 field emission gun scanning electron microscope (FEGSEM) (FEI/Thermofisher, Hillsboro, OR, USA), under high-vacuum conditions.

### 4.9. In Vitro Infection Studies

To further confirmed the synergistic activity of the complexes, the effect of HA–Lacto, HA–Que, HA–Lacto, and HA–Que-Lacto on human macrophage-like (U937) cell line, after 6 h of *Pseudomonas fluorescens* infection was carried out. The cells were maintained in RPMI supplemented with 10% FBS, 1% of Pen/strep, and cultured in 5% CO_2_ atmosphere. For infection, U937 cells were seeded onto 12-well plates at a density of 0.5 × 10^6^ cells/well, without antibiotics. Subsequently, the cells were infected with *Pseudomonas fluorescens* strains at a multiplicity of infection (MOI) of 100 with or without different treatment for 6 h. After the incubation period, the cells centrifuged for 5 min at 1500 rpm. The supernatant was discarded and the RNA was extracted using Trizol protocol [47]. A NanoDrop One/One^C^ Microvolume UV-Vis Spectrophotometer (Thermo Fisher Scientific Inc., Waltham, MA, USA) was used to assess total RNA quantity. The retro-transcriptase was performed in order to synthetize first-strand cDNA (SuperScript^®^ III Reverse Transcriptase, Invitrogen). Real-time PCR reactions were carried out in triplicate; expression values were calculated according to the 2^−∆∆Ct^ method [48], and all samples were normalized to *GAPDH* as a housekeeping gene. A negative sample (cell untreated) was used as calibrator.

## Figures and Tables

**Figure 1 ijms-22-09247-f001:**
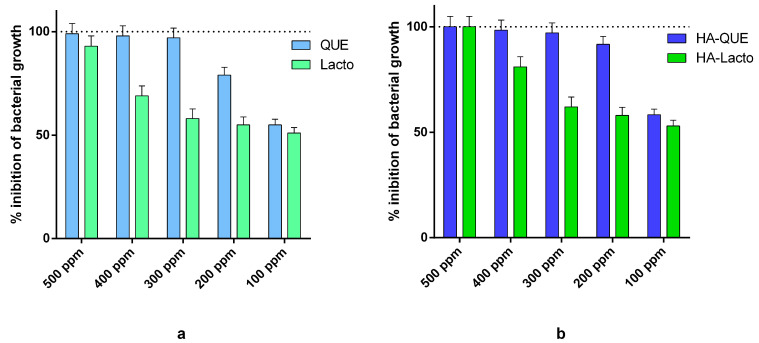
(**a**) Antimicrobial activity at different concentrations of Que and Lacto against *Pseudomonas fluorescens* (10^6^ CFU/mL). (**b**) Antimicrobial activity at different concentrations of HA–Que and HA–Lacto against Pseudomonas fluorescens (10^6^ CFU/mL).

**Figure 2 ijms-22-09247-f002:**
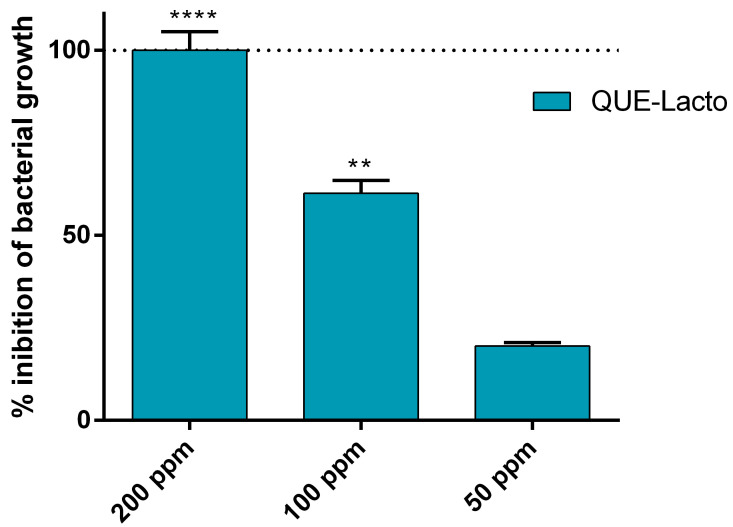
Antimicrobial activity of Que and Lacto together against *Pseudomonas fluorescens* (10^6^ CFU/mL). Statistical analysis was performed and considered statistically significant when *p* < 0.05 (** *p* < 0.01, **** *p* < 0.0001) according to two-way ANOVA multiple comparisons.

**Figure 3 ijms-22-09247-f003:**
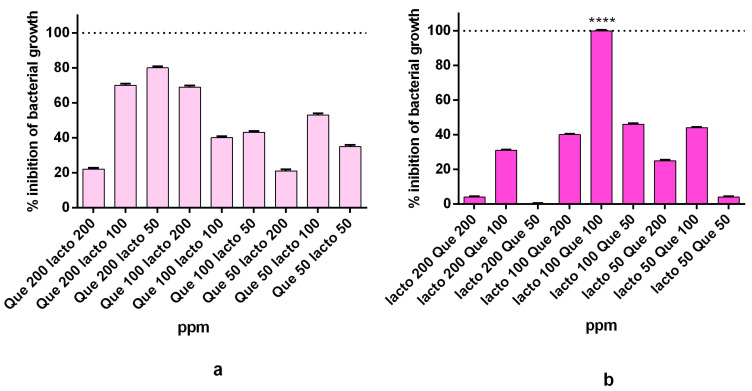
Antimicrobial activity of the Que and Lacto complex against *Pseudomonas fluorescens* (10^6^ CFU/mL). Different concentrations of Que and Lacto were absorbed on HA. The absorption order was Que and then Lacto (**a**) Lacto and then Que (**b**). Statistical analysis was performed and considered statistically significant when *p* < 0.05 (**** *p* < 0.0001) according to two-way ANOVA multiple comparisons.

**Figure 4 ijms-22-09247-f004:**
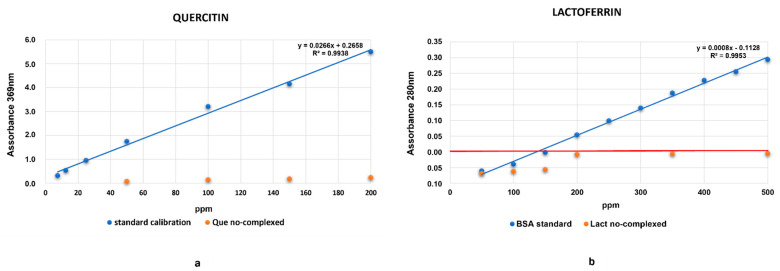
(**a**) The effective amount of Que (50–200 ppm) at the end of the adsorption process on HA. (**b**) The effective amount of Lacto (50–200 ppm) at the end of the adsorption process on HA.

**Figure 5 ijms-22-09247-f005:**
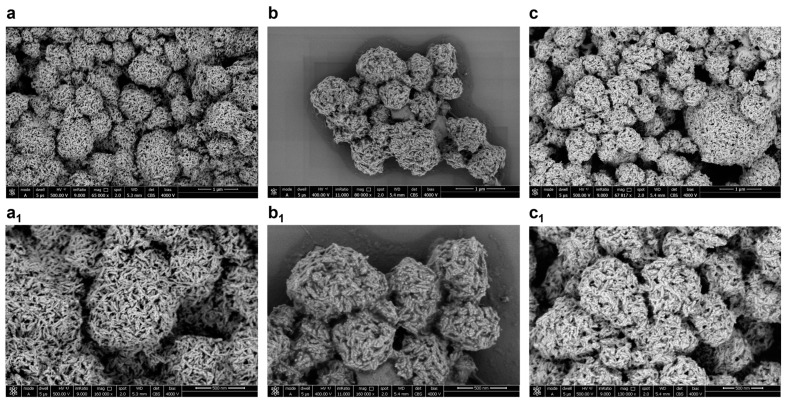
SEM image of: (**a**,**a_1_**) HA 65,000×, scale bar 1 µm, and 160,000×, scale bar 500 nm respectively; (**b**,**b_1_**) HA–Lacto 80,000×, scale bar 1 µm, and 160,000×, scale bar 500 nm respectively; (**c**,**c_1_**) HA–Lacto–Que 68,000×, scale bar 1 µm, and 130,000×, scale bar 500 nm respectively.

**Figure 6 ijms-22-09247-f006:**
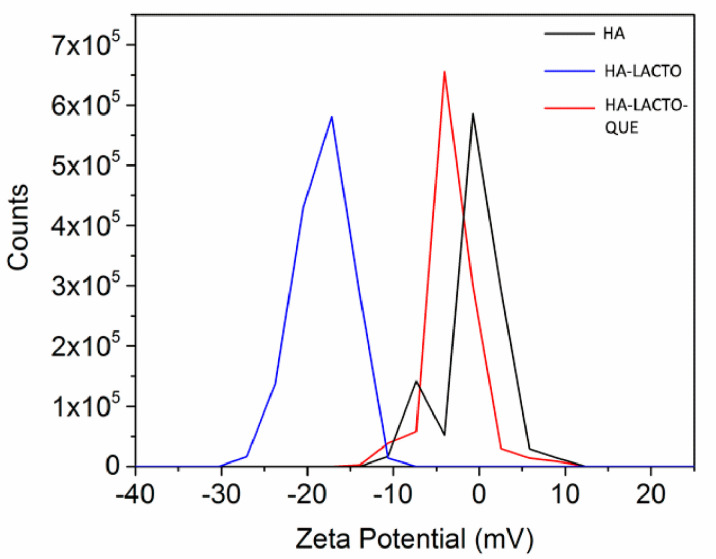
Zeta potential analysis of HA, HA–Lacto, and HA–Lacto–Que.

**Figure 7 ijms-22-09247-f007:**
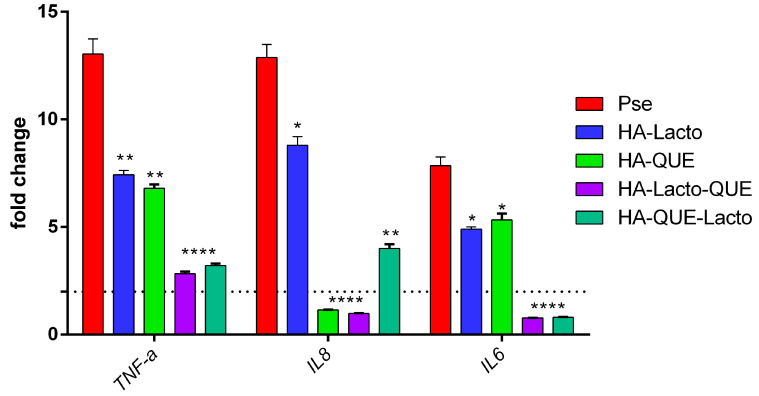
Real-time PCR cytokine expression profile of the different genes involved in the pro-inflammatory response of U937 cells in the presence of *Pseudomonas fluorescens* and the complexes for 6 h. Statistical analysis was performed and considered statistically significant when *p* < 0.05 (* *p* < 0.05, ** *p* < 0.01, **** *p* < 0.0001) according to two-way ANOVA multiple comparisons.

**Figure 8 ijms-22-09247-f008:**
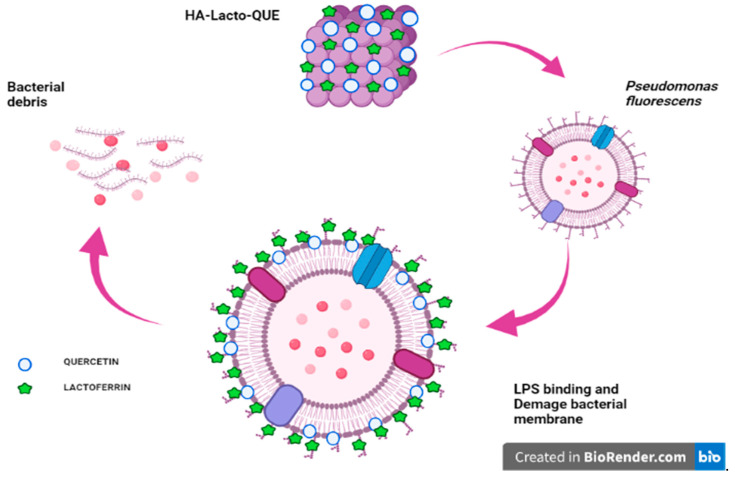
Schematic representation of the activity of HA–Lacto–Que complex against the cell wall of *Pseudomonas fluorescens*. The synergistic activity is due to the specific aptitude of each component: the affinity of Lacto for LPS and of Que for the bacterial membrane. Created with Biorender (https://biorender.com/, accessed on 4 August 2021).

**Table 1 ijms-22-09247-t001:** The values (*C_syn_*) are the result of the mathematical analysis used to determine the combination and concentrations of Que and lacto, with the best synergistic antimicrobial activity. The combination of HA–Lacto–Que (100 ppm *w*/*v*) showing the best synergistic *C_syn_* values.

	**Lacto 200**	**Lacto 100**	**Lacto 50**
**Que 100**	0.83 ± 0.18	1.27 ± 0.39	0.89 ± 0.21
**Que 100 + HA**	1.77 ± 0.50	0.88 ± 0.35	1.51 ± 0.52
	**Que 200**	**Que 100**	**Que 50**
**Lacto 100**	1.67 ± 0.33	0.85 ± 0.30	2.19 ± 0.44
**Lacto 100 + HA**	2.91 ± 0.77	6.75 ± 2.67	2.09 ± 0.66

## Data Availability

The data presented in this study are available within the article.

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
