# Peer review of "Lactoferrin, Quercetin, and Hydroxyapatite Act Synergistically against Pseudomonas fluorescens"

_ijms, 2021, doi:10.3390/ijms22179247_

Round 1

Reviewer 1 Report

The manuscript presented for review entitled "Lactoferrin, Quercetin and Hydroxyapatite act synergistically against Pseudomonas fluorescens" is in interesting experimental work.

It is written in accordance with the requirements for research work.

As a reviewer, I am asking for minor corrections:

line 56: Proteus spp. not Proteus

line 58. I would suggest increase the number of papers confirming the  activity of quercetin.

Fig 1. Pseudomonas fluorescens 106 Cfu/ml not Pseudomonas fluorescens 106cfu/ml

Fig 3, line 139,   - Italisized Pseudomonas fluorescens

Chapter 2.3.

I would use the FIC parameter to assess the synergistic effect.

e.g. Bidaud AL, Schwarz P, Herbreteau G, Dannaoui E. Techniques for the Assessment of In Vitro and In Vivo Antifungal Combinations. J Fungi (Basel). 2021;7(2):113. Published 2021 Feb 4.

After minor corrections are made, the manuscript can be published.

Author Response

Comments and Suggestions for Authors

The manuscript presented for review entitled "Lactoferrin, Quercetin and Hydroxyapatite act synergistically against Pseudomonas fluorescens" is in interesting experimental work.

It is written in accordance with the requirements for research work.

As a reviewer, I am asking for minor corrections:

  • line 56: Proteus spp. not Proteus

We modified the line 56 according to reviewer.

  • line 58. I would suggest increase the number of papers confirming the activity of quercetin.

As kindly suggested, new references have been added:

  1. Plaper A, Golob M, Hafner I, Oblak M, Solmajer T, Jerala R. Characterization of quercetin binding site on DNA gyrase. Biochem Biophys Res Commun. 2003.
  2. Lee JH, Park JH, Cho HS, Joo SW, Cho MH, Lee J. Anti-biofilm activities of quercetin and tannic acid against Staphylococcus aureus. Biofouling, 2013.
  • Fig 1. Pseudomonas fluorescens 106 Cfu/ml not Pseudomonas fluorescens 106cfu/ml

We modified the Figure 1 according to reviewer

  • Fig 3, line 139, - Italisized Pseudomonas fluorescens

We modified the Figure 3 according to reviewer

  • Chapter 2.3.

I would use the FIC parameter to assess the synergistic effect.

e.g. Bidaud AL, Schwarz P, Herbreteau G, Dannaoui E. Techniques for the Assessment of In Vitro and In Vivo Antifungal Combinations. J Fungi (Basel). 2021;7(2):113. Published 2021 Feb 4.

As Kindly suggested, we added the Fic parameter to assess the synergistic effect (Please see lines 138-141 and 285-289)

After minor corrections are made, the manuscript can be published.

Reviewer 2 Report

In the present manuscript, authors explained influence of  Lactoferrin, Quercetin and Hydroxyapatite against  Pseudomonas fluorescens.  Authors presented the study in an organized manner and results are presented very well. However, the graphs are not up to the journal quality. I recommend authors to re-draw graphs.

Author Response

In the present manuscript, authors explained influence of Lactoferrin, Quercetin and Hydroxyapatite against Pseudomonas fluorescens.  Authors presented the study in an organized manner and results are presented very well.

  • However, the graphs are not up to the journal quality. I recommend authors to re-draw graphs.

As kindly suggested, all the figures are modified according to the journal.